# Dementia from Small Vessel Disease Versus Alzheimer’s Disease: Separate Diseases or Distinct Manifestations of Cerebral Capillopathy Due to Blood–Brain Barrier Dysfunction? A Pilot Study

**DOI:** 10.3390/ijms26115040

**Published:** 2025-05-23

**Authors:** Charles R. Joseph, Davis A. Melin, Lindsay K. Wanner, Bryant Hartman, Jason Badelita, Lucy C. Conser, Harrison D. Kline, Pranav M. Pradhan, Kim Love

**Affiliations:** 1Department of Internal Medicine and Neurology, Liberty University College of Osteopathic Medicine, Lynchburg, VA 24502, USA; damelin@liberty.edu (D.A.M.); lkwanner@liberty.edu (L.K.W.); bhartman1@liberty.edu (B.H.); jbadelita@liberty.edu (J.B.); lcconser@liberty.edu (L.C.C.); hdkline@liberty.edu (H.D.K.); pmpradhan@liberty.edu (P.M.P.); 2K. R. Love Quantitative Consulting and Collaboration, Athens, GA 30605, USA; kim@krloveqcc.com

**Keywords:** 3D PASL MRI, arterial perfusion phase, capillary perfusion phase, Alzheimer disease, small vessel disease dementia, Montreal Cognitive Assessment, Fazekas scale, Koedam scale, Scheltens scale

## Abstract

Pathophysiological differences separating small vessel disease (SVD) from Alzheimer’s disease (AD) may alter treatment approach. Investigating peak-arterial and late-capillary perfusion may differentiate SVD from AD. 14 Subjects with MoCA scores of 11–24 were divided into 2 groups. Group one: 6 AD likely subjects positive for 1 or 2 copies of APOE 4+. Group two: 8 SVD likely subjects APOE−. Group three: 7 age-matched controls (MoCA 26–30). All underwent 3D PASL MRI, FLAIR, and SWI axial MRI. Arterial phase peak amplitude and latency, late capillary inflow/clearance rates, and anatomic abnormalities quantitated using microhemorrhage count, Fazekas, Koedam, and Schelton scales were compared. Arterial perfusion demonstrated no statistical differences among SVD, AD, and controls, suggesting normal arterial flow. Late phase perfusion showed significant localized reduction in capillary flow/clearance rates in SVD and AD compared to controls. Absent arterial phase but significant capillary inflow/clearance differences from controls suggest SVD and AD share common impaired blood–brain barrier origins.

## 1. Introduction

Dementia is a worldwide problem with a prevalence in 2019 of 55 million, with a care cost of $1.31 billion US (not to mention the enormous burden placed on families and caregivers), with estimates that by 2050, 152.8 million individuals will be affected [1,2]. The prevalence is greatest in the middle-income countries, with care costs greatest in the highly developed nations [2,3]. The two most common causes are Alzheimer’s disease (AD) (60–70%) and vascular dementia from small vessel disease (SVD) (20%) [1,3]. Cerebral small vessel disease-related dementia, excluding stroke, is on the basis of ischemia/hypoxia (arteriolar), BBB dysfunction, and rarely vasculitis [4]. Given the potential differences in therapeutic approach to both, clear knowledge of the pathologic underpinnings of each disease is essential [4,5,6,7,8,9]. Complicating distinction is the presence of ubiquitous SVD of greater or lesser degree in AD patients in both early and late-stage dementia [8,9,10].

Small vessel disease manifests as white matter hyperintensities (WMH) on fluid attenuated inversion recovery (FLAIR) MRI sequence, with or without a discrete lacunar infarction. WMH clinically not associated with typical stroke-like symptoms are found incidentally on brain MRI frequently in the upper age groups [11,12,13,14]. Capillary vascular anatomic changes have been well described in the literature in association with neurodegenerative disease [14,15,16,17,18]. These anatomic changes appear to increase with age: 9.5% of people ages 50–59 and 73% of people ages 80–89 were found to have cerebral SVD-related markers [14,15,16,17,18]. Vascular reorganization and atresia have been described in the SVD process as well as in patients with AD, contributing to the pathological burden [17]. Their pathologic origin may be of arteriolar occlusive disease origin or distinctly different and unrelated to the lipo-hyalinization causing discrete lacunar infarcts [5,9,13,14,19,20,21]. Alternatively, SVD purported origins may develop from blood–brain barrier (BBB) leakage or capillary basis, leading to their fibrosis occlusion and remodeling [12,13,20,21,22,23,24,25]. Risk factors for developing either stroke or SVD are similar, e.g., hypertension, smoking, alcohol abuse, obesity, and diabetes [26,27,28]. By exploring early phase arterial and late phase capillary perfusion in cognitively impaired subjects, one group WMH+ APOE 4+ (genetically disposed for early AD) versus WMH+ APOE—subjects not predisposed for early AD, may distinguish arterial physiological differences between the two groups [12,29,30,31] [Figure 1]. If so, then the underpinnings of SVD are distinct from the protein misfolding in AD. Alternatively, if no difference in the arterial phase or the late phase of perfusion is found between the two groups, but similar abnormalities in the capillary phase, then a single origin for both should be considered. Ascertaining SVD origin will affect future approaches to early management of AD [28,29,30,31,32].

Three-dimensional PASL MRI using endogenous spin-labeled blood is suited for non-invasive determination of flow by measuring the signal present within a defined region of interest at multiple post-labeling delays (PLD’s) [34]. Reliability of ASL MRI in imaging arteriolar perfusion deficits is lower given the delayed transit times related to low signal to noise (S/N) and partial volume, particularly if only a single PLD is acquired [35]. By comparing peak amplitude and latency during the early perfusion phase, the peak arterial inflow is measured [34,35,36,37,38]. With arteriopathy, a reduction in peak amplitude with delayed latency compared to normal would be expected [35,36,37]. If present, SVD would be distinguished from AD, where arterioles are spared [29,38]. If on the other hand, the arterioles are spared in SVD, then the early arterial phase of perfusion should not be affected [38,39,40].

By using the same method but choosing late post-labeling delay times (PLD), the capillary inflow/outflow rate can be ascertained [36,37,41,42]. If capillaries are damaged, then the late phase of perfusion should show reduced clearance rate of capillary blood flow when either group is compared to controls [38,39,40,41,42,43]. Potential correlation of arterial flow abnormalities in brain regions demonstrating late perfusion abnormalities would add evidence of arteriopathy in SVD if not seen in AD [34]. To that end, this study addresses the presence or absence of arteriopathy in early dementia presumed AD (APOE 4+) with probable SVD (APOE 4−) and comparing both to age-matched normal controls.

To further compare the two groups with each other and controls, anatomic abnormalities using FLAIR MRI to quantify WMH with the Fazekas scale, degree of temporal atrophy, Schelton scale, and degree of Parietal lobe atrophy by Koedam scale were employed. The number of microhemorrhages present on Susceptibility Weighted Imaging (SWI MRI) was counted and compared.

## 2. Results

### 2.1. Summary of Data

The 22 variables [Table 1] are reported using mean and minimum/maximum, as initial inspection using visualizations and Shapiro-Wilk tests showed that these variables are frequently skewed/not normally distributed within one or more of these groups.

Data analysis was performed using IBM SPSS Statistics v. 30. Statistical significance was set at α = 0.05.

### 2.2. Arterial Phase Comparisons

Box and whisker plots comparing peak latency and peak amplitude values across the three groups are illustrated in Figure 2 and Figure 3. The results of the Kruskal-Wallis tests showed there were no statistically significant differences in arterial phase peak latency or amplitude values for any region across the three groups (see Appendix A
Table A1 for data).

### 2.3. Glymphatic Slope Comparisons

Late perfusion phase glymphatic slopes within the six brain regions greater than −0.02 are considered to indicate abnormal clearance found only in SVD and AD subjects. All clearance rates in the control group for all brain regions were less than 0.02 see [Figure 4].

Box and whisker plots comparing glymphatic slope values across the three groups are illustrated in [Figure 5] (see Appendix A
Table A2 for data).

Kruskal-Wallis tests indicate statistically significant differences between the groups in all regions except for the right parietal and left front (Appendix A Table A3). Post-hoc comparisons are varied, but all significant differences show that SVD or AD slopes are greater than the normative group. There are no regions where the SVD and AD groups are significantly different from one another.

**Figure 4 ijms-26-05040-f004:**
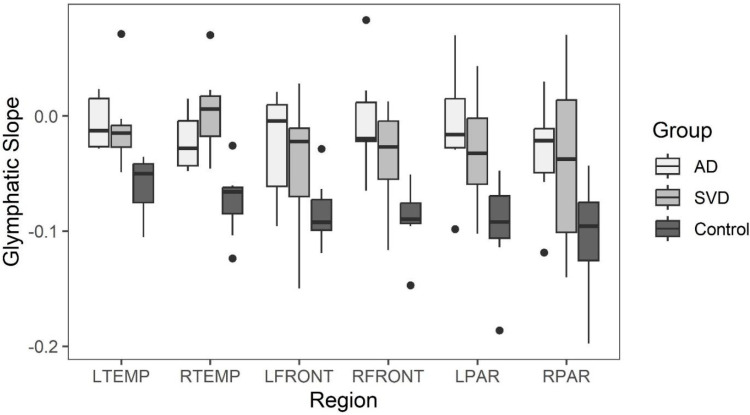
Late phase glymphatic slopes for AD, SVD, and control subjects by region. Significant differences in glymphatic clearance slopes are noted when comparing SVD and AD slopes with the controls. There are no regions where the SVD and AD groups are significantly different from one another.

Peak amplitude and peak latency from the arterial phase of perfusion for brain regions with abnormal late glymphatic phase flow are compared across the three groups. The results of Kruskal-Wallis tests showed there were also no statistically significant differences in arterial phase peak latency or amplitude values for any region across the three groups for this reduced data (Appendix A
Table A4 and Table A5).

Summary whisker graphs based on this selection of abnormal regions of glymphatic flow are illustrated in Figure 5 and Figure 6. Sample sizes are provided separately for each region (see Appendix A
Table A6 for data).

**Figure 5 ijms-26-05040-f005:**
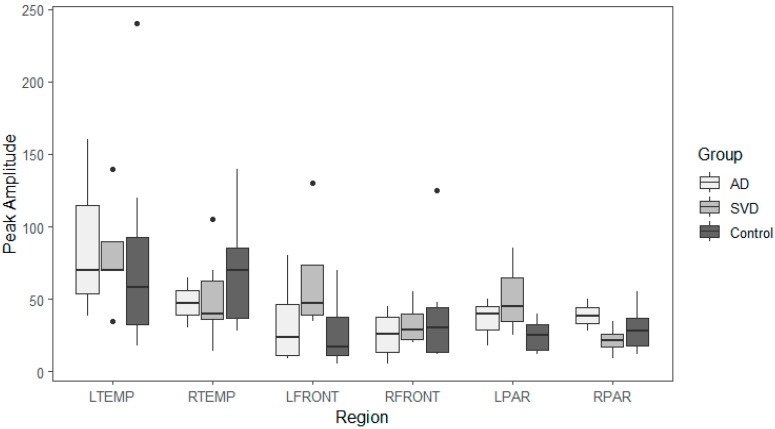
Arterial phase peak amplitude for AD, SVD, and control subjects by region. Here, AD and SVD subjects were compared to each other and to controls, including data limited to each specific region with reduced glymphatic clearance. No significant differences between the groups AD, SVD, and controls were noted. Thus, the arterial phase amplitudes were unimpaired.

**Figure 6 ijms-26-05040-f006:**
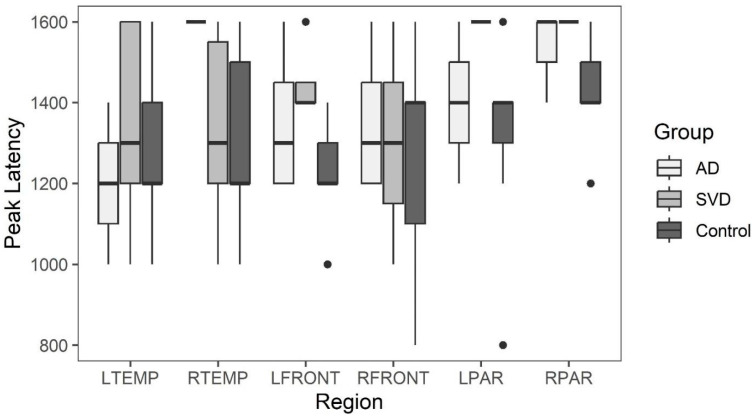
Arterial phase peak latency for AD, SVD, and control subjects by region. Here, for AD and SVD subjects, the included data have been limited to each specific region with reduced glymphatic clearance. No significant differences between SVD and AD, or when either group is compared to controls in the arterial phase in brain regions demonstrating significant reduction in glymphatic phase clearance. The arterial phase peak latency was unimpaired in those regions.

### 2.4. Montreal Cognitive Assessment (MOCA) and Memory Index Score (MIS) 

MoCA and its subtest, MIS, scores were approximately normally distributed within each group with the mean and SD [Table 2].

These scores were compared between the AD and SVD groups using a Welch’s *t*-test, which allows for different variances in the two groups. The result for MOCA scores was t(11.7) = −1.76, *p* = 0.105; the result for MIS scores was t(10.8) = −3.49, *p* = 0.005. There is no evidence that the MOCA scores of the two groups are different; the MIS scores in the SVD group are statistically significantly higher than those in the AD group.

### 2.5. MRI Image Score Comparisons

The results of Kruskal-Wallis tests comparing scale scores (Fazekas, Koedam, and Scheltens) and the count of microhemorrhages across the three groups are provided in Table 3 (see Appendix A
Table A7 for data set). There were no statistically significant differences in the Koedam and microhemorrhages across the groups; however, the Fazekas scale score was significantly higher in the SVD group than the normative group (*p* = 0.012), and the Scheltens scale score was significantly higher in the AD group than the normative group (*p* = 0.006). No significant differences were found between the AD and SVD groups in either the Fazekas scale or the Scheltens scale.

## 3. Discussion

The results did not support the hypothesis that arterial perfusion differences are present between the APOE 4− (SVD) and the APOE 4+ (AD) group. Further, neither group differed in the arterial phase of perfusion compared to controls. Although compelling, the absence of differences may be related to sample size and genetic diagnostic assumptions separating the two groups. There is precedent literature supporting the absence of arterial perfusion abnormality in SVD with evidence of capillopathy [44]. There was a trend in comparing the memory index scores (MIS) of short-term memory loss in the AD group compared with the SVD group. The latter suggests accentuated temporal lobe involvement in the AD group, as might be suggested with earlier β-amyloid accumulation and its toxic effect on these highly metabolically active structures [45].

Limitations: A larger study in the early dementia stage will further clarify any distinction between SVD and AD. Amyloid PET scan could provide additional evidence for AD. That said, one subject in the SVD group had a negative PET obtained via a second opinion at another medical institution 8 months after their study MRI. Absence or presence of β-amyloid plaque in the early disease (MCI) stage does not rule out/in the diagnosis of AD [46,47,48,49]. Finally, this study purposely excluded individuals with cortical stroke from large or small vessel arterial occlusion, as the mechanism of disease is clearly different.

On the other hand, both groups (SVD and AD) demonstrated focal regions of statistically significant reduced capillary inflow/outflow (late perfusion phase) compared to controls. This speaks to delayed capillary perfusion time and retained labeled fluid within the ROI, both consequences of BBB dysfunction affecting both groups. Additionally, no statistically significant anatomical differences were found between the SVD and AD groups. Both showed statistically significant anatomic differences from normal controls. Thus, from a physiologic and anatomic perspective, there are no distinguishing differences between the two groups. The co-occurrence of both pathologies, specifically capillary perfusion dysfunction and accumulating misfolded proteins, may share the same root origin but develop via different mechanisms.

The cerebral capillary system is more complex and extensive than any other within the body, comprising 80% of the cerebral vasculature [50]. The endothelial cells in particular are tasked with selective active and passive transport of essential small and large molecules via specific small molecule transporters and selective protein transport by both carrier-mediated transport and receptor-mediated transcytosis [51]. They are integral in establishing the glymphatic channels for transport out of waste [52]. In addition, they provide the expression of tight junction proteins between cells, thus establishing the BBB [53]. Employing endothelial transport mechanisms and barrier maintenance is via astroglia indirectly through pericyte signaling, the components dubbed the neurovascular unit (NVU) [51]. The BBB is under constant assault via the inflammatory influences produced by the multiple risk factors mentioned above, which increase with age [54,55,56]. The well-known result is a slow leak of the BBB with potential destructive consequences for both capillary integrity and altered metabolic synthetic, and degradation pathways [56].

The presence of SVD has varied effects on cognition depending on its extent, location, and progression over time [57]. The high-risk factors linked to SVD are the usual suspects: hypertension, diabetes, smoking, obesity, multiple head traumas, senescent cells, alcohol abuse, chronic infection, and newer elucidated risk factors of anxiety, depression, and sleep apnea. All of these factors are also associated with a high risk of late-life AD [58,59,60,61].

The effects of enhanced inflammatory signaling via circulating cytokines have been shown to upregulate microglial complement secretion with resulting phenotypic alteration in microglia, converting them to proinflammatory cells [62]. Via microglial C3a upregulating endothelial cell C3r, converting them to immune attractant cells with retraction of tight junctional proteins, causing resultant BBB leak and dysregulated endothelial transport [63]. Polarization of aquaporin four channels in astrocyte end feet also occurs, thus shutting down the glymphatic pathway [64]. The result is further intrusion of restricted proteins such as fibrinogen with secondary expression of glial inflammatory pathways affecting both capillary integrity and thus flow, as well as altering β-amyloid cleavage pathways and later in the process of hyperphosphorylation Tau proteins [64]. These downstream events slowly unfold, albeit by different pathways, leading to capillary injury/loss and accumulation of toxic misfolded proteins [65,66,67]. Hereditary factors accelerate the early accumulation of misfolded proteins, which in turn accelerates secondary collateral BBB damage in a vicious cycle leading to cognitive decline [68].

In a previous study of mild head injury in young athletes showing acute reduced capillary/glymphatic flow followed by recovery of normal flow correlating with clinical improvement, two questions arose. By what mechanism is the BBB restored? Why does that natural recovery pathway diminish with age [67,69]? The gradual BBB leak with age is well recognized [45,70,71,72]. Before the questions can be answered, the natural mechanism of repair must be ascertained. Previous studies have shown statistically significant capillary/glymphatic flow differences in patients with diagnosed AD from normal controls [37,43,73,74]. Additionally, the temporal progression of cerebral capillopathy, as demonstrated by progressive decline in late phase perfusion in focal brain regions has been shown in subjects with MCI [38].

It is clear from all the various treatment trials that effective management to halt the neurodegenerative disease process must be provided in the early or preclinical phase of the disease. Removing the misfolded protein byproduct of the disrupted metabolic processes has limited benefit. Perhaps it could be likened to ‘removing dead fish from a polluted pond and expecting the pollution to clear when the source thereof is upstream’ [38]. If the two processes share the early common capillary pathology with detrimental leak of normally restricted substances, then future novel therapeutic efforts to restore BBB integrity in the preclinical stage may prevent dementia [75]. Until the nexus leading to both the capillopathy and toxic proteinopathy is addressed, namely the persistent BBB leak, the disease process will likely progress. The co-occurrence of both processes compellingly suggests a common initiating disease process as well. The slow age-related leak of the blood–brain barrier induced by the presence of various inflammatory triggers appears most likely, with the cascade of destructive pathways disrupting the capillary conduits and the NVU cellular elements [Figure 7].

Approaching this disease from its apparent nexus point, BBB leak, may bear substantial fruit therapeutically. The question that remains is how do we identify individuals in the preclinical phase of disease, and then, given the current state of the art, reduce late-life risk of dementia? Who and how do we screen patients? What is the most cost-effective and least invasive means of doing so? The latter questions have yet to be answered. The target cohort is those with significant associated validated risk factors for late-life dementia. Patients with hypertension, diabetes, cigarette abuse, obesity, multiple head injuries, chronic infections/inflammation, hyperlipidemia, chronic anxiety/depression, chronic alcohol abuse, or family history of dementia are the most reasonable targets for screening [76,77]. Treatment in those identified with preclinical disease by virtue of validated screening measures requires intensified therapy of modifiable risk factors with follow-up physiologic testing [78]. The targeting of hypertension, reduced cigarette abuse, and careful diabetic management has in fact reduced the incidence of dementia in developed countries; however, the potential degree the obesity epidemic has on truncating that trend is yet to be determined [79]. When novel therapeutic means are available to upregulate the natural BBB repair mechanisms that either become faulty or overwhelmed by age-related risk factors, the neurodegenerative disease process may be averted. The role of extracting accumulating misfolded proteins (dead fish) may certainly remain part of the management plan as well.

**Figure 7 ijms-26-05040-f007:**
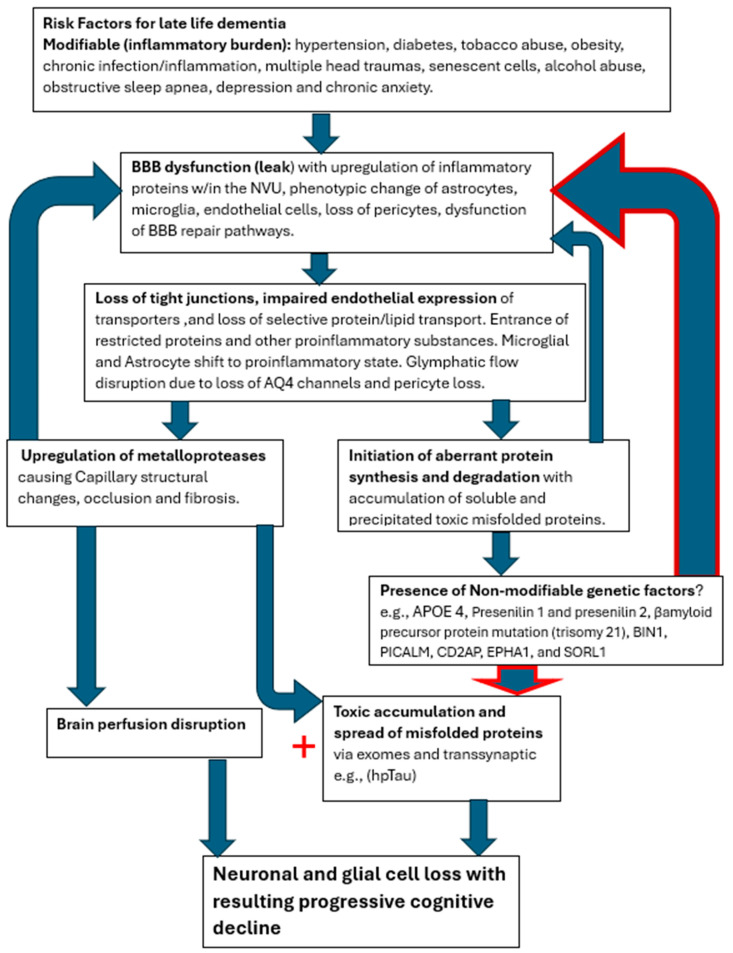
Cerebral capillopathy schematic leading to Alzheimer’s dementia, including risk factors, initial target (BBB dysfunction with leak), producing vascular injury, and metabolic consequences. Accelerating factors that further capillary dysfunction include accumulating misfolded proteins, capillary loss, and genetic factors (GF). The rate of cognitive decline increased by GF, reducing the normal handling of misfolded proteins.

## 4. Methods

This pilot study was approved by the CENTRA Health institutional IRB CHIRB0605. All methods were performed in accordance with the relevant guidelines and regulations.

Pilot study size estimate was uncertain as estimates of how strong these effects are unknown as to the presence or absence of arterial phase differences among the groups. The number of subjects required to demonstrate statistically significant differences in MCI and dementia from controls in the capillary or late phase determination was thus used. 

Six APOE 4 positive subjects with mild dementia/MCI (MoCA 11–24) were recruited along with 8 APOE 4 negative subjects with SVD and mild dementia/MCI from the CENTRA Health memory disorders clinic. All subjects were between 55–85 years of age and recruited from August 2023 to June 2024. An additional seven age-matched normal controls (MoCA 26–30) were also studied. Exclusion criteria included symptomatic ischemic stroke (subject 6 in the SVD group had a prior cerebellar hemorrhage), heart disease, or other organ failure, systemic vascular disease, claustrophobia, and MRI incompatible implants. Informed consent was obtained from all subjects. No financial remuneration was given.

MRI studies were obtained in the late afternoon in all subjects on a 3T Skyra Siemens MRI machine (Erlangen, Germany). They were advised to withhold caffeine for the prior 6 h (standard MRI protocol) and fast for the prior 2 h. All subjects had the MoCA study within 3 months of the MRI study. All sequences were transferred to DICOM for processing and evaluation. Total scan time was 26 min 40 s.

The MRI protocol included: 3D PASL (early capillary phase) 5 sequences, 3D PASL (late capillary/glymphatic) phase 7 sequences, FLAIR axial sequence, and SWI (susceptibility weighted imaging) sequence (see Appendix B for sequence details).

The resulting 3D PASL images were reformatted into contiguous 4 mm axial slices using standard slice angles. They were transferred along with the Flair and SWI MRI images to DICOM for analysis.

All axial PASL images were evaluated at bilateral homologous temporal, frontal, and parietal anatomic levels for all subjects. Hand-drawn standardized ROI’s of standard volumes for each brain region for both early and late perfusion analysis, avoiding subarachnoid and ventricular CSF spaces were obtained, transferred, and catalogued by brain region to PowerPoint. Temporal ROI volumes analyzed were 650 mm^3^, frontal lobe volumes 1150 mm^3^, and parietal volumes 760 mm^3^ (Figure 8). Resulting signal averages for each PLD and location were then graphed on an Excel Spreadsheet. Best fit linear analysis was applied to the 7 PLD’s (per anatomic location) in late perfusion (capillary) phase [37]. The slope of the line is the capillary inflow/clearance rate of signal per time [37]. The 5 PLD’s in arterial (early) perfusion phases were graphed using non-linear analysis, and the maximum peak amplitude and its latency were recorded for all six anatomic locations [42].

Late phase capillary clearance rates and early phase arterial peak amplitude and latencies were compared among all three groups (Table 1). Additionally, the arterial phase results corresponding to brain regions showing reduced capillary (late-phase) clearance rates (defined as having slopes greater than −0.02) were compared to arterial phase results in unaffected brain regions [37,43,44,45].

White matter hyperintensities identified on the FLAIR MRI sequence for all subject groups were clinically graded using the Fazekas scale (Table 1). The presence of angular gyrus parietal atrophy was graded using the Koedam scale, and temporal lobe atrophy was graded using the Scheltens scale [46]. Direct count of microhemorrhages on SWI was also performed (Table 2). All MoCA and MIS results were compared between the AD and SVD groups using an independent *t*-test.

## 5. Conclusions

Our study supports the presence of cerebral capillopathy in both subjects with early dementia, APOE 4− (SVD) and APOE 4+ (AD probable). The two groups were not separable from each other or from controls on the basis of ASL MRI arterial flow abnormalities. Both shared similar late capillary/glymphatic phase dysfunction. Given their near-universal combined appearance in neurodegenerative disease, a common origin related to BBB dysfunction/leak initiating both downstream pathologies is implied. Further research into the pathophysiology of BBB dysfunction, and in particular repair pathways diminished with age, may provide the needed silver bullet for effective early treatment.

## Figures and Tables

**Figure 1 ijms-26-05040-f001:**
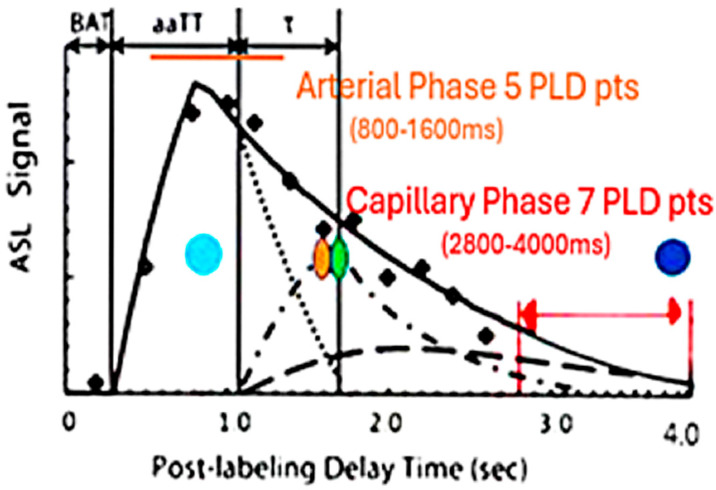
Perfusion cycle with arterial, venous, and capillary components versus time. The 5-arterial phase PLD times are illustrated by the orange line (800–1200 ms, 200 ms intervals), the 7-capillary phase PLD’s (2800–4000 ms at 200 ms intervals) by the red line. T1 times (63% signal decay) of major tissue signal contributors are indicated by the colored dots. Magenta dot = 800–850 ms T1 of white matter. Orange dot = 1650 ms T1 of blood. Green dot = 1700 ms T1 of gray matter (all values are for 3T). Dk blue dot = 3800 ms T1 of water (CSF fluid); (all values are for 3T). BAT = bolus arrival time; aaTT =artery-artery arrival time; τ = peak capillary arrival time. Adapted with publisher permission (Wiley) from [33].

**Figure 2 ijms-26-05040-f002:**
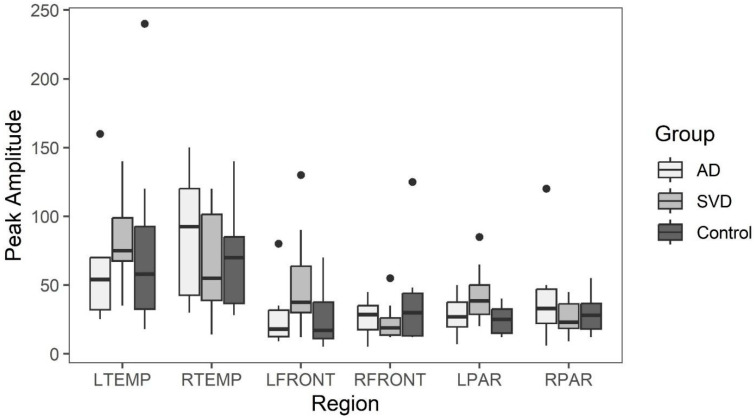
Early phase arterial peak amplitude for AD, SVD, and control subjects by region. AD = Alzheimer’s Disease, SVD = small vessel disease, L/RTEMP = Left/Right temporal lobe, L/RFRONT = L/R Frontal lobe, L/RPAR = L/R Parietal lobe. There were no significant statistical differences among the three groups. The box =interquartile range (IQR) 25 (1st) to 75 (3rd) percentile; whisker = 1st IQR − 1.5 X1st IQR and 3rd IQR + 1.5X 3rd IQR. Black dots = outliers in line with their respective group. This applies to Figure 3, Figure 4, Figure 5 and Figure 6.

**Figure 3 ijms-26-05040-f003:**
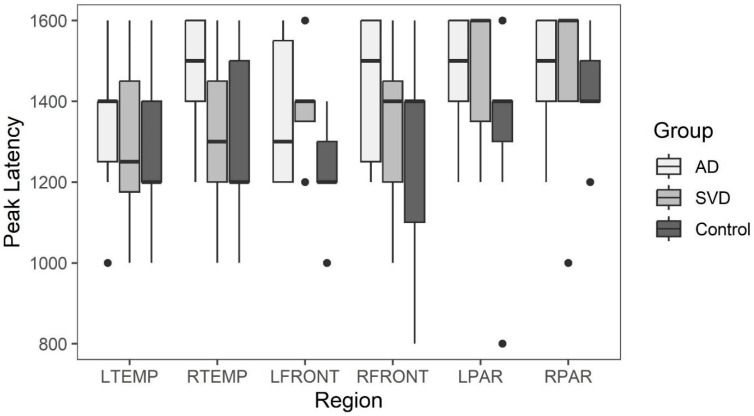
Early phase arterial peak latency for AD, SVD, and control subjects by region. There were no significant statistical differences among the three groups.

**Figure 8 ijms-26-05040-f008:**
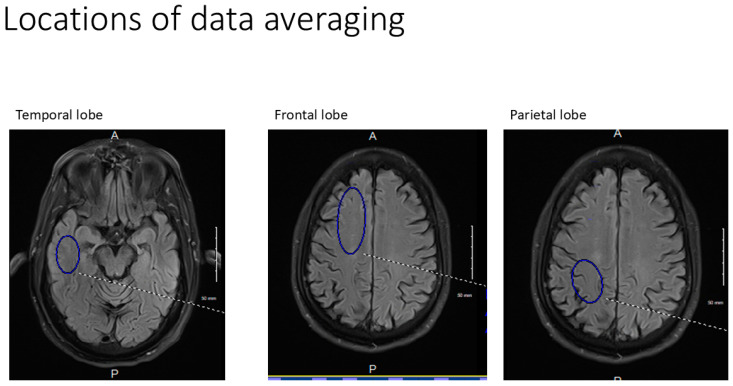
Note the locations of the Region of Interest drawn for each anatomic location (shown as blue ellipses unilaterally for clarity). Temporal lobe volume = 650 mm^2^, Frontal lobe = 1150 mm^2^, Parietal lobe = 760 mm^2^.

**Table 1 ijms-26-05040-t001:** Compendium of demographic (age MoCA scores, and SVD and AD APOE status), 0–17: Severe cognitive impairment, 18–25: Mild cognitive impairment (MCI), 26–30: Normal cognitive function. Memory index score (MIS) 0–6: Severe memory impairment, 7–11: Mild memory impairment, 12–15: Normal memory function. ASL MRI results amplitude and peak latency in the Arterial perfusion phase in the six brain regions studied, slopes of the clearance rates in the capillary/glymphatic late phase of perfusion in the same six anatomic regions with the identical size of region of interest. The arterial phase values in red are correlated with corresponding significantly abnormal capillary phase values also in red. The presence of white matter hyperintensities on the FLAIR sequence was visually quantified using the Fazekas scale. The presence of parietal atrophy was scored using the Koedam scale, the presence of temporal lobe atrophy was scored with Schelton’s scale, and the number of microhemorrhages found on SWI was directly counted. PL = peak latency in milliseconds, PA = peak amplitude in arbitrary units, AD = subjects APOE 4 positive for one or two copies with MCI or mild dementia, SVD = small vessel disease with MCI or mild dementia, APOE = Apolipoprotein E, MoCA = Montreal Cognitive Assessment, MIS = Memory Index score, NT = not tested, FSS = Fazekas scale score, KS = Koedam score, SSTL = Scheltens Scale Temporal Lobe, MH = microhemorrhages. Table 1 provides the values of the variables analyzed in this report. There were 6 AD individuals, 8 SVD individuals, and 7 control individuals included in this data set; individuals had values reported for 22 different variables (peak latency, peak amplitude, and glymphatic slope for each of 6 different regions, plus the Fazekas scale score, Koedam score, Scheltens’ scale, and microhemorrhage count). MoCA = Montreal Cognitive Assessment, version 8.1 Copyright @ 2024 MoCA Test Inc. All rights reserved.

				Arterial Peak Phase Latency and Amplitude												Glymphatic Phase Slope									
				LTEMP		RTEMP		LFRONT				LPAR		RPAR											
Group	APOE 4	MOCA	MIS	PL	PA	PL	PA	PL	PA	PL	PA	PL	PA	PL	PA	LTEMP	RTEMP	LFRONT	RFRONT	LPAR	RPAR	FSS	KS	SSTL	MH
AD	POS	16/30	2/15	1400	25	1400	35	1200	9	1200	5	1600	40	1200	6	−0.0267	−0.0478	−0.0128	−0.0194	−0.0097	−0.0248	2	0	1	0
AD	POS	16/30	3/15	1200	70	1400	120	1200	14	1400	16	1200	18	1400	20	0.0199	−0.0480	−0.0957	−0.0198	0.0233	−0.1186	2	1	2	0
AD	POS	13/30	5/15	1400	70	1600	150	1200	80	1200	35	1400	50	1600	120	−0.0268	−0.0295	0.0210	0.0834	0.0701	−0.0574	3	2	4	2
AD	POS	16/30	3/15	1400	38	1600	65	1600	22	1600	22	1600	24	1600	38	0.0234	0.0031	−0.0772	−0.0649	−0.0984	−0.0180	1	1	2	0
AD	POS	18/30	6/15	1000	160	1200	120	1600	35	1600	45	1400	7	1400	28	0.0012	−0.0266	0.0041	0.0221	−0.0293	−0.0087	1	0	2	0
AD	POS	11/30	5/15	1600	30	1600	30	1400	12	1600	35	1600	30	1600	50	−0.0283	0.0149	0.0116	−0.0227	−0.0226	0.0299	2	2	3	3
SVD	NEG	14/30	4/15	1200	90	1200	100	1400	130	1400	55	1400	65	1400	45	−0.0191	−0.0315	−0.0128	−0.0194	−0.0893	−0.0930	2	1	2	0
SVD	NEG	21/30	8/15	1400	80	1400	35	1200	90	1000	35	1600	85	1400	40	−0.0413	0.0099	−0.0544	0.0126	0.0434	−0.1258	3	2	1	0
SVD	NEG	24/30	10/15	1000	140	1400	120	1200	35	1200	14	1600	45	1000	14	−0.0100	−0.0457	−0.1173	−0.1165	−0.1023	−0.1402	2	0	1	1
SVD	NEG	21/30	11/15	1600	70	1200	14	1400	12	1200	20	1200	30	1600	20	0.0714	0.0154	−0.0279	0.0105	−0.0423	0.0485	3	1	1	2
SVD	NEG	15/30	5/15	1600	70	1600	70	1400	35	1400	18	1600	25	1600	23	−0.0026	−0.0131	−0.0043	−0.0529	−0.0026	−0.0131	2	2	2	2
SVD	NEG	21/30	11/15	1200	60	1200	105	1400	40	1400	12	1200	20	1600	9	−0.0222	0.0704	−0.0164	−0.0611	−0.0222	0.0704	2	0	2	3
SVD	NEG	16/30	12/15	1100	125	1000	40	1400	15	1600	12	1600	32	1600	35	−0.0490	0.0024	−0.1500	−0.0343	−0.0492	0.0024	2	1	2	0
SVD	NEG	13/30	6/15	1300	35	1600	40	1600	55	1600	23	1600	45	1600	23	−0.0107	0.0226	0.0281	−0.0095	−0.0003	−0.0617	2	0	3	4
N	NT	28/30	13/15	1200	58	1200	43	1000	17	1400	13	1600	40	1600	28	−0.0618	−0.0663	−0.0285	−0.0509	−0.0477	−0.0432	1	1	1	0
N	NT	28/30	14/15	1000	240	1200	90	1200	5	1600	13	1400	18	1600	33	−0.0463	−0.0258	−0.0821	−0.0712	−0.0983	−0.0825	2	1	1	0
N	NT	26/30	10/15	1200	65	1200	80	1200	70	1000	125	1400	25	1400	40	−0.0890	−0.1238	−0.0922	−0.0800	−0.1864	−0.1977	2	0	1	0
N	NT	28/30	13/15	1600	40	1600	70	1200	30	1400	40	1400	12	1200	18	−0.0500	−0.0659	−0.1058	−0.0911	−0.0709	−0.1141	1	0	0	0
N	NT	29/30	14/15	1600	18	1400	30	1400	10	1200	12	800	30	1400	12	−0.0356	−0.0605	−0.1191	−0.1472	−0.0681	−0.1370	0	0	0	0
N	NT	28/30	11/15	1200	25	1600	28	1400	12	800	30	1400	12	1400	18	−0.0370	−0.0636	−0.0926	−0.0897	−0.0920	−0.0679	0	0	1	0
N	NT	30/30	15/15	1200	120	1000	140	1200	45	1400	48	1200	35	1400	55	−0.1053	−0.1037	−0.0634	−0.0956	−0.1142	−0.0957	1	0	1	2

**Table 2 ijms-26-05040-t002:** Summary of MoCA and MIS scores and standard deviations for the SVD with AD group scores compared to controls.

	MoCA		MIS	
	Mean	SD	Mean	SD
**AD**	15.0	2.53	4.0	1.55
**SVD**	18.1	4.08	8.4	3.06
**Control**	28.1	1.22	12.86	1.77

**Table 3 ijms-26-05040-t003:** Results of Kruskal-Wallis Tests comparing MRI image scores among the three groups which demonstrate a significant difference only between the Fazekas score (number of hyperintensities on FLAIR sequence) between the SVD and normal groups but not between the SVD and AD group. Likewise, the Scheltens scale showed a significant difference between the AD and normal groups but not between the SVD and AD groups.

Scale	χ^2^	df	*p*	Post-Hoc Results
Fazekas Scale Score (Numeric estimate of white matter hyperintensities)	8.33	2	0.016	SVD > N (*p* = 0.012)
Koedam Score (Mesial parietal atrophy)	3.10	2	0.213	n/a
Scheltens’ Scale (Temporal lobe atrophy)	10.39	2	0.006	AD > N (*p* = 0.006)
Microhemorrhages	3.51	2	0.173	n/a

## Data Availability

All data from this study are presented in Table 2 and Appendix B. MRI images are available by request from the corresponding author.

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
