# Peer review of "Dementia from Small Vessel Disease Versus Alzheimer’s Disease: Separate Diseases or Distinct Manifestations of Cerebral Capillopathy Due to Blood–Brain Barrier Dysfunction? A Pilot Study"

_ijms, 2025, doi:10.3390/ijms26115040_

Round 1

Reviewer 1 Report

Comments and Suggestions for Authors

The study supports the presence of cerebral capillopathy in both subjects with early dementia APOE 4- (SVD) and APOE 4+ (probable AD).

The experimental results that support the conclusions of this paper are shown in supplementary tables.

The methodology used is clear, detailed, and comprehensively described.

Table 1 contains illegible data in the printed version; however, it can obviously be enlarged for analysis in the electronic version.

The discussion of the results is extensively presented in figures and tables.

Figure 8 clearly shows the cerebral capillaropathy schematic leading to Alzheimer's dementia, including risk factors, initial target (BBB dysfunction with leak) producing vascular injury, and metabolic consequences.

The references are extensive, adequate, and up to date.

This is good work worthy of publication in this journal.

Author Response

The study supports the presence of cerebral capillopathy in both subjects with early dementia APOE 4- (SVD) and APOE 4+ (probable AD).

The experimental results that support the conclusions of this paper are shown in supplementary tables.

The methodology used is clear, detailed, and comprehensively described.

Table 1 contains illegible data in the printed version; however, it can obviously be enlarged for analysis in the electronic version.

The discussion of the results is extensively presented in figures and tables.

Figure 8 clearly shows the cerebral capillaropathy schematic leading to Alzheimer's dementia, including risk factors, initial target (BBB dysfunction with leak) producing vascular injury, and metabolic consequences.

The references are extensive, adequate, and up to date.

This is good work worthy of publication in this journal.

Reviewer 2 Report

Comments and Suggestions for Authors

Dementia is one of a major public health problems. Besides health conditions it generates vast costs of treatment and care for those affected and the society. It should be mentioned also that it causes enormous social burden for patients and carers. Most of the cases of dementia are Alzheimer’s (AD) cases, it constitutes about 60 or 70% of all dementia cases.

Joseph and colleagues present a pilot study comparing status of small blood vessels (cerebrovascular) between (SVD -small vessels disease and (AD) versus control group without mild dementia. What they found is that capillary flow/clearance rate in both representing disease groups SVD and AD is impaired to the same extent and smaller than in control group. The authors conclude that it is possible that both diseases have common origin in dysfunction of BBB (brain blood barrier) however with distinct manifestations. This idea is in line with those theories which are seeking origins of AD in aging processes occurring much earlier than presence of amyloid pathology. The main points of this idea are presented very clearly in Fig. 8. The problem with this manuscript is fact that authors build very elaborated theory based on data obtained on relatively small number of cases. However, it is a pilot study and results are very interesting and warrant larger study. Therefore, I recommend acceptance of this manuscript after minor revision.

Minor points,

1. I wonder what means “metabolic misfolding” in line 50. I would expect ether “proteinopathies” or “protein misfolding”

2. This is rather question to the authors - do CAA (cerebral amyloid angiopathy) occurs in SVD?. Is it possible to check it by PET with contrasting compounds?

Author Response

Dementia is one of a major public health problems. Besides health conditions it generates vast costs of treatment and care for those affected and the society. It should be mentioned also that it causes enormous social burden for patients and carers. Most of the cases of dementia are Alzheimer’s (AD) cases, it constitutes about 60 or 70% of all dementia cases.

Joseph and colleagues present a pilot study comparing status of small blood vessels (cerebrovascular) between (SVD -small vessels disease and (AD) versus control group without mild dementia. What they found is that capillary flow/clearance rate in both representing disease groups SVD and AD is impaired to the same extent and smaller than in control group. The authors conclude that it is possible that both diseases have common origin in dysfunction of BBB (brain blood barrier) however with distinct manifestations. This idea is in line with those theories which are seeking origins of AD in aging processes occurring much earlier than presence of amyloid pathology. The main points of this idea are presented very clearly in Fig. 8. The problem with this manuscript is fact that authors build very elaborated theory based on data obtained on relatively small number of cases. However, it is a pilot study and results are very interesting and warrant larger study. Therefore, I recommend acceptance of this manuscript after minor revision.

Minor points,

1. I wonder what means “metabolic misfolding” in line 50. I would expect ether “proteinopathies” or “protein misfolding”

2. This is rather question to the authors - do CAA (cerebral amyloid angiopathy) occurs in SVD?. Is it possible to check it by PET with contrasting compounds?

Reviewer 3 Report

Comments and Suggestions for Authors

1. In this study, the patients were classified into AD and SVD groups, but there is a discrepancy with the definitive diagnosis of AD or SVD based on clinical diagnosis. The additional criteria for AD or SVD diagnosis should be made clearer, and if possible, the accuracy of the disease type classification should be ensured by supplementary tests, or this limitation should be emphasized more in the Discussion.

2. The technical details of the MRI parameters were somewhat complicated and burdensome for the reader. We recommend that the details of the MRI protocol be organized and placed in the Supplementary Material or Table, and only an outline is provided in the main text.

3. We would like the visibility of the figures and the supplementary interpretation to be improved (especially Figs. 5–7).
The reader's understanding would be deepened by visually indicating significant differences using "*" or "ns" in the figures.

Author Response

  1. In this study, the patients were classified into AD and SVD groups, but there is a discrepancy with the definitive diagnosis of AD or SVD based on clinical diagnosis. The additional criteria for AD or SVD diagnosis should be made clearer, and if possible, the accuracy of the disease type classification should be ensured by supplementary tests, or this limitation should be emphasized more in the Discussion.
  2. The technical details of the MRI parameters were somewhat complicated and burdensome for the reader. We recommend that the details of the MRI protocol be organized and placed in the Supplementary Material or Table, and only an outline is provided in the main text.
  3.  
